# Horizontal Cervix as a Novel Sign for Predicting Adhesions on the Posterior Extrauterine Wall in Cases of Placenta Previa

**DOI:** 10.3390/jcm8122141

**Published:** 2019-12-04

**Authors:** Shinya Matsuzaki, Aiko Okada, Masayuki Endo, Yoshikazu Nagase, Satoshi Nakagawa, Kosuke Hiramatsu, Aiko Kakigano, Kazuya Mimura, Tsuyoshi Takiuchi, Takuji Tomimatsu, Yutaka Ueda, Kazuhide Ogita, Tadashi Kimura

**Affiliations:** 1Department of Obstetrics and Gynecology, Osaka University Graduate School of Medicine, 2-2 Yamadaoka, Suita, Osaka 565-0871, Japan; aikoriver@yahoo.co.jp (A.O.); endo@gyne.med.osaka-u.ac.jp (M.E.); doctoryoshikazu@gmail.com (Y.N.); s.nakagawa@gyne.med.osaka-u.ac.jp (S.N.); hiramatwo_shiwasu@yahoo.co.jp (K.H.); mimura@gyne.med.osaka-u.ac.jp (K.M.); takkitakkitakki@gyne.med.osaka-u.ac.jp (T.T.); tomimatsu@gyne.med.osaka-u.ac.jp (T.T.); y.ueda@gyne.med.osaka-u.ac.jp (Y.U.); tadashi@gyne.med.osaka-u.ac.jp (T.K.); 2Department of Obstetrics and Gynecology, Aizenbashi Hospital, Osaka 556-0005, Japan; 3Department of Health Science, Osaka University Graduate School of Medicine, 2-2 Yamadaoka, Suita, Osaka 565-0871, Japan; 4Department of Obstetrics and Gynecology, Rinku General Medical Center, Osaka 598-0048, Japan; k-ogita@rgmc.izumisano.osaka.jp

**Keywords:** placenta previa, endometriosis, magnetic resonance imaging

## Abstract

We aimed to identify a magnetic resonance imaging (MRI) feature that can predict posterior extrauterine adhesion (posterior adhesion) antenatally, in patients with placenta previa. We retrospectively reviewed patients with placenta previa who underwent a preoperative MRI examination of placenta accreta spectrum. We categorized the patients into two groups based on whether the cervix was anterior or posterior to a line perpendicular to the anatomical conjugate on the MRI. We projected the perpendicular line toward a straight line through the broad of the back on T2-weighted sagittal MRI images and measured the angle between this line and the line passing through the cervical canal. We analyzed the correlation of the cervical canal angle with the presence of posterior adhesions. Of the 96 patients analyzed, 71 patients had an anteverted cervix and 25 patients had a retroverted cervix. There were 21 posterior adhesions. The adhesion rate was significantly higher in patients with a retroverted cervix than those with an anteverted cervix (8.5% vs. 60%; *p* = 0.00). The cervical canal angle was ≤10° in 25 patients; of these 17 had adhesions (sensitivity, 81.0%; specificity, 89.3%; area under the curve, 0.887; 95% confidence interval, 0.792–0.981). This finding, labeled “positive horizontal cervix sign,” may be a promising indicator of posterior adhesions in patients with placenta previa.

## 1. Introduction

Placenta previa is associated with significant intraoperative bleeding during cesarean delivery [1]. Placenta previa complicated by placenta accreta spectrum (PAS) poses a higher risk for massive hemorrhage [2,3]. The risk of placenta previa is higher in patients with endometriosis than in those without endometriosis (6% vs. 1%) as reported in a systematic review [4,5]. The development of assisted reproductive technologies has increased the rates of pregnancy in patients with endometriosis [6], which in turn might increase the number of pregnancies complicated by placenta previa.

Although endometriosis might be a major cause for posterior extrauterine wall adhesions (posterior adhesions) during pregnancy, other factors such as repetitive pelvic inflammatory disease and scars from a previous pelvic surgery also increase the risk of posterior adhesions [7,8]. Various methods, such as uterine compression suture, vertical compression sutures of the lower uterine segment, hemostatic square sutures, and hysterectomy, have been performed to control bleeding in patients with placenta previa or PAS [9,10,11]; however, these methods require exteriorization of the uterus, and posterior adhesions might increase surgical morbidity. To the best of our knowledge, no methods to evaluate pregnant patients for posterior adhesions have been reported.

Magnetic resonance imaging (MRI) was performed to assess PAS in select cases of placenta previa [12,13,14]. Our preliminary analysis indicated that the cervical canal was posterior to a line perpendicular to the anatomical conjugate in a patient in whom posterior adhesions were confirmed intraoperatively. Hence, we retrospectively analyzed MRI images to investigate the association between the direction of the cervical canal and the presence of posterior adhesions.

## 2. Materials and Methods

We retrospectively reviewed medical records from Osaka University and the Rinku General Medical Center. We included data from patients with placenta previa (including low-lying placenta) who underwent MRI for the preoperative assessment of PAS, who delivered via cesarean delivery, and whose surgical records were available for examination between January 2008 and June 2016. This study was approved by the Osaka University Ethics Committee (Approval #16046, approved on 22 July 2016).

Informed consent was not obtained from the patients because of the retrospective nature of this study based on computerized data with anonymous selection, which did not subject the patients to new interventions. As an alternative to written informed consent from the patients, an opt-out document was created which contained information about the study design and publication of the results. This document was posted on our institution’s website to provide patients with an opportunity to halt the disclosure of their medical records. None of the patients refused to provide data. Therefore, the need for written informed consent was waived by the Ethics Committee that approved this study’s protocol.

We recorded the clinical characteristics, including age, parity, gestational age at MRI examination, number of previous cesarean deliveries, body mass index, neonatal birth weight, in vitro fertilization (IVF) status, location of the placenta, gestational age at delivery, operative procedure, and postoperative histological examination for PAS. Patients were classified into the following four groups according to gestational age at the time of MRI examination: group 1 (≤26 weeks), group 2 (between 27 and 29 weeks), group 3 (between 30 and 32 weeks), and group 4 (≥33 weeks or later).

MRI scans of all patients included in our study were performed at Osaka University or the Rinku General Medical Center using 1.5-T superconducting magnets manufactured by General Electric (GE Healthcare, Waukesha, WI, USA), Philips (Philips North America, Andover, MA, USA), or Siemens (Siemens Corporation, Washington, DC, USA), using rapid acquisition T1- and T2-weighted pulse sequences. During the first image analysis, we examined whether the cervical canal was located anterior or posterior to a line perpendicular to the anatomical conjugate line (Figure 1). Thus, we first defined the anatomical conjugate line (line ①). Next, we drew a dotted line perpendicular to the line on the internal os (line ②). We then recorded the position of the cervix as either anteverted (Figure 1a) or retroverted (Figure 1b) in relation to the dotted line.

After collecting these data from all patients, we assessed them to determine an association between the position of the cervix and the presence of a posterior adhesion. Next, we calculated the cervical canal angle. We projected a straight line through the broad of the back (Figure 2a, line A) on T2-weighted sagittal images. By drawing a line passing through the internal os perpendicular to line A (Figure 2a, line B), we identified the line passing through the internal os to the external os (Figure 2a, line C). We used a protractor to measure the angle between lines B and C (Figure 2a, lines B and C) and defined it as the cervical canal angle (Figure 2a, angle D). A typical image of the cervical canal angle of 0° is shown in Figure 2b and we defined this angle as the reference.

We measured the cervical canal angle position as positive if the measured angle was clockwise and increased with respect to the reference angle position, and negative if the measured angle was anticlockwise and decreased with respect to the reference angle position (Figure 2c). A typical image of the cervical canal angle measured as −10° is shown in Figure 2c. We rounded up the measurements, with the minimum unit defined as 5°. If the value could not be divided as integers, the value was increased (e.g., 15°/2 = 7.5° and 7.5° was increased to 10.0° because the minimum unit was defined as 5°). In all cases, two obstetricians, unaware of the patients’ clinical characteristics and surgical findings, independently measured the angle. We used the mean values for our analysis.

In practice, the surgeons performed exteriorization of the uterus for repair of hysterotomy during the cesarean delivery for placenta previa and recorded the presence or absence of posterior adhesions in the surgical records. In our study, we considered posterior adhesions to be present if any of the following was documented: (1) adhesions between the posterior extrauterine wall and the small bowel, colon, rectum, ovary, or pelvic wall; (2) exteriorization of the uterus was impossible because of the posterior adhesions; (3) adhesions were dissected for extracorporeal elevation; and (4) intraoperative bleeding occurred from the posterior extrauterine wall because of the posterior adhesions.

In cases where extraperitoneal elevation was impossible, the surgeons recorded the reason as the presence of a posterior adhesion. We investigated the presence of posterior adhesions by assessing the surgical records. We excluded adhesions between the abdominal wall and the anterior uterine wall, and those between the bladder and the anterior uterine wall. We classified patients into an adhesion group and a non-adhesion group based on the presence/absence of posterior extrauterine adhesions.

The primary aim of this study was to identify an MRI finding that can predict posterior adhesions. The secondary aim of this study was to assess the ideal angle of the cervical canal to predict posterior adhesion. We used a linear regression model with a scatter plot diagram created using the EZR software (Saitama Medical Center, Jichi Medical University, Saitama, Japan) to compare variables between the adhesion and non-adhesion groups [15]. Categorical variables were analyzed using the Mann–Whitney U test, chi-square test, and Fisher’s exact test. All statistical analyses were performed using JMP Pro 12.2 (SAS Institute, Cary, NC, USA). Significance was set at a two-sided *p*-value of <0.05.

No patients were involved in setting the research question or the outcome measures, nor were they involved in developing plans for recruitment, design, or implementation of the study. No patients were asked to advise on interpretation or writing up of results.

## 3. Results

Our study included data from 96 patients who were divided into two groups according to the direction of the cervix. Table 1 shows the clinical characteristics of these patients. In all, 71 out of 96 patients had an anteverted cervix and 25 patients had a retroverted cervix. There were significantly more primiparas (26.8% vs. 72%; *p* < 0.01) and the rate of a previous cesarean delivery was significantly lower in the retroverted cervix group (50.7% vs. 24%; *p* < 0.01). Placenta previa with posterior placenta was more frequent in the retroverted cervix group compared with the anteverted cervix group (31% vs. 76%; *p* < 0.01).

Out of the 71 patients with an anteverted cervix, six patients presented with posterior adhesions, and of the 25 patients with a retroverted cervix, 15 patients presented with posterior adhesions. According to these results, the rate of posterior adhesions was significantly higher in patients with a retroverted cervix than in those with an anteverted cervix (8.5% vs. 60%; *p* = 0.00).

We calculated the reliability of retroverted cervix to predict posterior uterine adhesions and found the sensitivity to be 71%, specificity to be 87%, positive predictive value (PPV) to be 60%, and negative predictive value (NPV) to be 92%. Posterior adhesions were more frequent in patients with a retroverted cervix (*p* < 0.01). The typical MRI findings for cases with and without posterior adhesions are shown in parts d and e of Figure 2, respectively.

We found that the mean cervical canal angle was significantly smaller in the adhesion group than in the non-adhesion group (5.95° vs. 38.41°; *p* < 0.01). We measured the cervical canal angle in the non-adhesion group to evaluate the physiological changes according to gestational age and found that the mean cervical canal angles were 64.4° in group 1, 42.8° in group 2, 36.4° in group 3, and 28.4° in group 4 (Figure 3).

To investigate the relationship between the presence of placenta accreta spectrum (PAS) and cervical canal angle, the cervical canal angle in PAS cases was determined and compared with those in non-PAS cases in groups 2 and 3 (Figure 4). No significant changes in gestational weeks at MRI examination (29.6 weeks vs. 29.8 weeks; *p* = 0.82) and cervical canal angle were observed (36.1° vs. 43.4°; *p* = 0.14) between the non-PAS and PAS groups. These results suggested that the cervical canal angle was not associated with the presence of PAS.

The statistical analysis of continuous variables of the cervical canal angle indicated that the cervical canal angle tended to decrease by 3.3° per week as the gestational age increased (Figure 5; *p* < 0.01, linear regression analysis). The mean cervical canal angle was 5.95° in the adhesion group, which increased by approximately 0.19° per week as the gestational age increased (*p* = 0.001). Figure 5 presents a scatter plot of the cervical canal angle and gestational age on MRI for cases with and without posterior adhesions.

An receiver operating characteristic (ROC) curve analysis determined the optimal cervical canal angle to detect posterior adhesion as 10° [sensitivity, 81.0%; specificity, 89.3%; PPV, 68%; NPV, 94.3%; area under the curve (AUC), 0.887; 95% confidence interval (CI), 0.792–0.981] (Figure 6a, Table 2). Our results suggested that a cervical canal angle of ≤10° was an abnormal sign. Therefore, we designated cases wherein the angle was ≤10° as having a positive “horizontal cervix sign.” Out of the 96 cases, 25 had positive cervical horizontal sign, and of these, 17 had posterior adhesions (Table 2).

When we analyzed the 65 patients who underwent MRI at a gestational age of ≤32 weeks, we found higher reliability of the horizontal cervix sign to predict posterior adhesions, with a sensitivity of 92.0%, specificity of 86.7%, PPV of 76.0%, NPV of 96.0%, AUC of 0.916, and 95% CI, 0.809–1.0 (Figure 6b, Table 2). If a cervical canal angle of ≤0° was designated as an abnormal sign (easier to detect), then this sign was used to predict posterior adhesions with a sensitivity of 67%, specificity of 100%, PPV of 100%, and NPV of 91% (Table 2). Although the ROC curve analysis found the optimal angle to detect posterior adhesion as 10°, a cervical canal angle of ≤0° at a gestational age of ≤32 weeks showed high PPV (100%), and we believe that this will easier to determine (an example image is shown in Figure 2e).

## 4. Discussion

A key finding of our study is that measurements of the cervical position could provide additional information regarding posterior adhesions. We noted that posterior adhesions were more frequent in cases with a retroverted cervix, and a cervical canal angle of ≤10° showed a high sensitivity and specificity for the detection of posterior adhesions.

Due to its high cost, MRI is performed in cases with a high risk for PAS. Our findings enable the detection of posterior adhesions without additional costs. Various methods mentioned above have been reported to control bleeding in patients with placenta previa or PAS [9,10,11]; however, these methods require exteriorization of the uterus. Furthermore, if a patient requires peripartum hysterectomy, then posterior adhesions might increase surgical morbidity. Our findings should be extremely useful for facilitating the identification of patients with posterior adhesions who are at risk for a difficult surgery owing to placenta previa or PAS.

Saraswat et al. compared late pregnancy outcomes in 4232 women with endometriosis and 6707 without endometriosis and reported that the rate of placenta previa was 1.7% in the former group and 0.8% in the latter (adjusted odds ratio, 2.2; 95% CI, 1.5–3.3) [16,17]. Other studies have also reported that the incidence of placenta previa is high in patients with endometriosis [4,5,18]; thus, the frequency at which obstetricians encounter placenta previa or PAS may increase with an increase in the incidence of endometriosis.

According to a previous study, endometriosis is found in 1–15% of women of reproductive age and is one of the most common causes of posterior adhesion in non-pregnant women [19,20,21]. Although endometriosis might be a major cause for posterior adhesion during pregnancy, the pathogenesis of this condition during pregnancy remains unknown. Other risk factors for posterior adhesion include infections, such as pelvic inflammatory disease, and a history of pelvic surgery [7,8]. Regardless of the cause, to the best of our knowledge, no diagnostic imaging technique is currently available for determining the presence of posterior adhesions during pregnancy.

MRI findings that suggest the presence of a posterior adhesion in non-pregnant women include uterine retroversion, elevated posterior fornix of the vagina, fibrosis of the serous surface of the pelvic wall, and intestinal adhesions [22,23]. We speculate that these findings might not be useful in pregnant women due to the difficulty in assessing an enlarged uterus during pregnancy; however, this has not been investigated to date. The ability to assess posterior adhesions in pregnant women at high risk of bleeding, such as those with placenta previa and PAS, is extremely useful because it allows early determination of the potential risks and planning of treatment strategies.

To the best of our knowledge, no previous study has investigated the association between the direction of the cervix and the presence of posterior adhesions. We showed that by measuring this angle, we could identify the presence of posterior adhesions with a high sensitivity (81.0%). In a secondary analysis, we determined that in cases without posterior adhesion, the cervical canal angle was approximately 64° at gestational week ≤26 and approximately 28° at gestational week ≥33. We found that the cervical canal angle decreased significantly by approximately 3.3° per week as the pregnancy advanced. We think that these findings reflect physiological changes, such as enlargement of the uterus and descent of the fetal head. In patients with posterior adhesions, we speculated that the cervical canal angle could not change due to the adhesions. We believe that the physiological change in the angle of the cervical canal on MRI has not been previously reported; thus, we believe that our finding is novel.

Considering the changes in cervical canal angle during pregnancy, when we limited our examination to gestational week ≤32, the results were much better. These data suggested that better predictions may be achieved if MRI examinations were performed by the 32nd gestational week. Particularly, cases with a cervical canal angle of ≤0° at gestational week ≤32 were always complicated by posterior adhesions (PPV, 100%).

Despite the high PPV of the positive horizontal sign, factors (e.g., fetal weight, twin pregnancies, uterine myoma, and uterine adenomyosis) that affected or biased the cervical angle changes could not be determined. To enhance the robustness of our findings, a prospective study with a larger number of cases should be conducted to investigate the effects of estimated fetal weight and twin pregnancies on the degree of cervical canal angle and the effects of the presence or absence of uterine myoma or adenomyosis.

The strength of our study is that it is the first study to focus on the cervical canal angle on MRI to assess posterior adhesions, which were found in approximately 80% of cases when the horizontal cervix sign was positive. If MRI examinations to assess PAS have already been performed, the horizontal cervix sign can help to identify the presence of posterior adhesions without any additional cost or burden to patients.

We are aware of the several limitations of our study. First, this study had a retrospective design, a relatively small sample size, and lacked longitudinal analysis; therefore, unmeasured bias may exist in the analysis. To accurately determine changes in the cervical canal angle during pregnancy, the cervical canal angle should be measured in each case at various gestational ages. Serial MRI examination, however, could not be performed due to its high cost. Therefore, this may have caused bias in our data. In addition, factors affecting and causing bias on the degree of cervical canal angle should also be investigated.

Second, surgical dictations of posterior adhesion may be very non-specific, thus representing more bias in this study. To overcome ambiguity, we have shown the intraoperative images in Appendix A. To improve this point, an evaluation method for the severity of extrauterine adhesions should be established. With all this information, examining the association between cervical canal angle and severity of extrauterine adhesions may be useful. Further studies should be performed to confirm and expand our findings.

Third, the causes of posterior adhesion were not analyzed. The majority of posterior adhesions have been considered to be caused by endometriosis; however, this could not be proven with histopathological analysis. Fourth, to clearly show the accuracy of our findings, a more robust study should be conducted. Furthermore, whether identifying the horizontal cervix sign improves the surgical outcome also remains to be elucidated. A prospective study seems appropriate because a randomized control study is difficult.

Fifth, MRI was not easily performed to investigate posterior adhesions because of its high cost. To resolve this problem, a future investigation is necessary to identify the horizontal sign using a transvaginal ultrasonography (TV-US). Serial TV-US will allow us to perform the longitudinal analysis of the cervical canal angle according to the gestational age and analyze each case at various gestational ages. With this information, we will investigate the cervical canal angle using TV-US to detect posterior extrauterine adhesions in the future study.

In conclusion, the horizontal cervix sign may be promising as a predictor of posterior adhesions in patients with placenta previa.

## Figures and Tables

**Figure 1 jcm-08-02141-f001:**
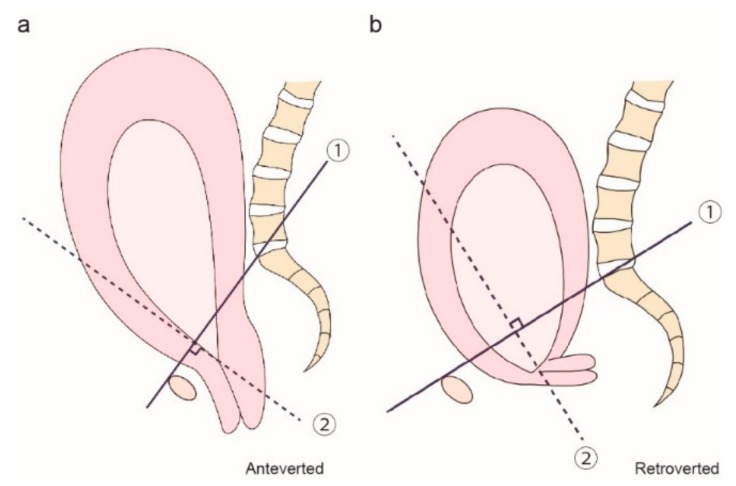
Method for evaluating the position of the cervix as either anteverted or retroverted. Line ① indicates the anatomical conjugate and line ② indicates the line perpendicular to the anatomical conjugate. We defined an anteverted cervix (**a**) as that anterior to line ② and a retroverted cervix (**b**) as that posterior to line ②.

**Figure 2 jcm-08-02141-f002:**
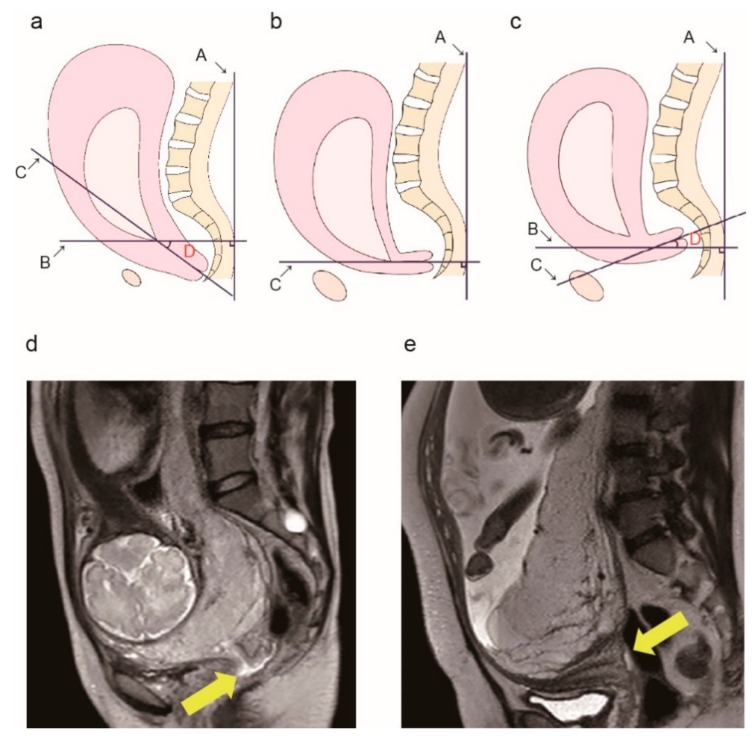
Measurement of the cervical canal angle. (**a**) Diagram showing the method used to measure the cervical canal angle. Line A: A straight line through the broad of the back. Line B: A line perpendicular to line A. Line C: A line passing through the internal os to the external os (cervical canal line). D: The angle formed by lines B and C (defined as the cervical canal angle). (**b**) The cervical canal angle is 0° because lines A and C are perpendicular to each other. We defined this angle as the reference. (**c**) A sample image of a positive horizontal sign. Cervical canal is at a 10° angle counterclockwise along line B, thus the cervical canal angle is −10°. Typical magnetic resonance imaging (MRI) findings in a patient without posterior extrauterine wall adhesion (**d**) and a patient with posterior extrauterine wall adhesion (**e**). Yellow arrows indicate the cervical canal.

**Figure 3 jcm-08-02141-f003:**
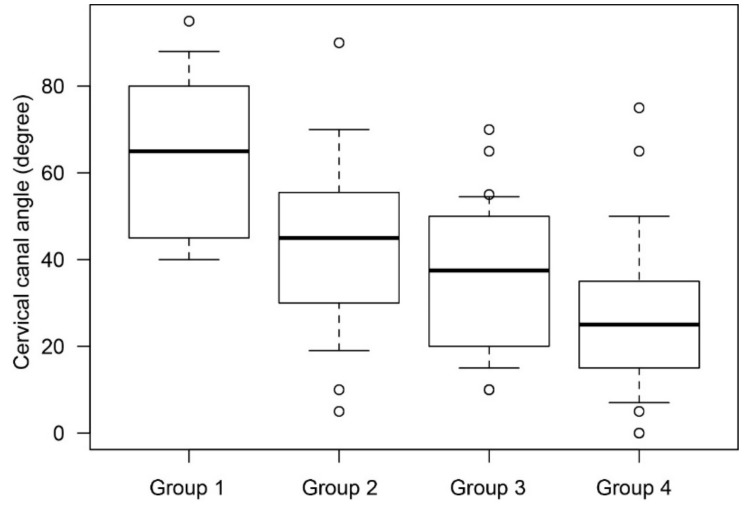
The mean cervical canal angle in cases without posterior extrauterine wall adhesion. The cervical canal angle was analyzed to evaluate physiological changes in the cervical canal according to gestational age at MRI. In this analysis, cases with posterior adhesion were excluded. The vertical axis presents the cervical canal angle and the horizontal axis presents the patient groups divided according to gestational age on MRI. Mean cervical canal angles were 64.4° in group 1, 42.8° in group 2, 36.4° in group 3, and 28.4° in group 4.

**Figure 4 jcm-08-02141-f004:**
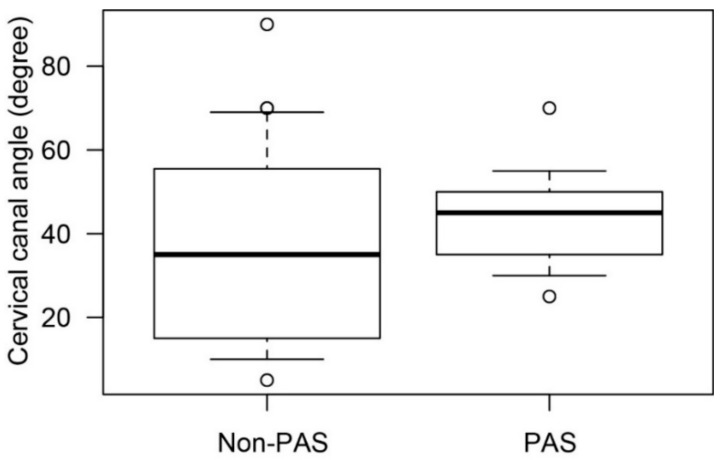
The mean cervical canal angle in cases with and without placenta accreta spectrum. To compare the cervical canal angle in patients with and without PAS in groups 2 and 3. In this analysis, cases with posterior adhesions were excluded. The mean cervical canal angle was 36.1° in non-PAS cases and 43.4° in PAS cases, and no significant difference was observed (*p* = 0.14). Abbreviation: PAS, placenta accreta spectrum.

**Figure 5 jcm-08-02141-f005:**
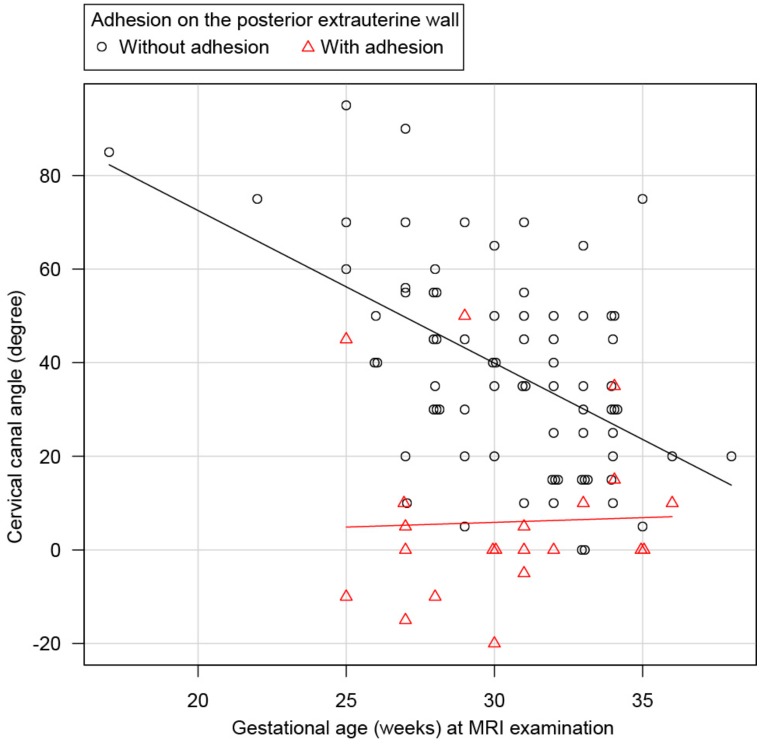
The statistical analysis of continuous variables of the cervical canal angle. A scatter plot diagram of the data obtained from 96 pregnant women who underwent MRI. The vertical axis presents the cervical canal angle and the horizontal axis presents the number of gestational weeks at MRI examination. ○ represents patients without posterior extrauterine wall adhesion and △ represents patients with posterior extrauterine wall adhesion. The approximate lines for the two groups of patients are indicated. The approximate line for the non-adhesion group showed that the cervical canal angle significantly decreased by 3.3° per week, whereas the approximate line for the adhesion group was 0°, indicating no significant change.

**Figure 6 jcm-08-02141-f006:**
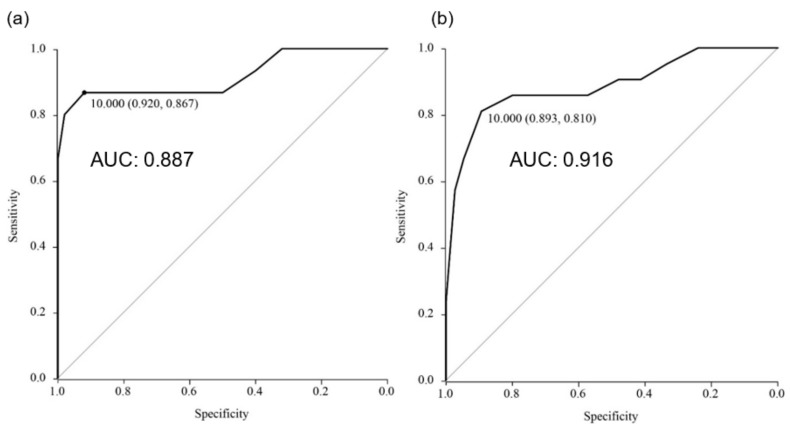
(**a**) ROC curve of cervical canal angle for all patients. (**b**) ROC curve of cervical canal angle investigated before 32 weeks of gestation.

**Table 1 jcm-08-02141-t001:** Patient baseline characteristics.

	All	Anteverted	Retroverted	
Number	96	71	25	*p*
Age (years)	35.23 ± 5.05	35.08 ± 4.68	35.7 ± 6.0	0.62
35 or older	55 (57.3%)	43 (60.6%)	12 (48.0%)	0.35
Primipara	37 (38.5%)	19 (26.8%)	18 (72.0%)	0.0001
Maternal BMI ^a^ (range)	23.5 ± 3.5 (17.4–41.0)	23.6 ± 3.4 (17.4–41.0)	23.2 ± 2.3 (19.4–27.8)	0.42
Birth weight ^b^	2560.1 ± 392.9 (1742–3770)	2581 ± 384.9 (1742–3770)	2500.3 ± 420.3 (1750–3160)	0.31
Number of previous CD				
0	54 (56.2%)	35 (49.3%)	19 (76.0%)	0.035
1	21 (21.9%)	16 (22.5%)	5 (20.0%)	0.58
2 or more	21 (21.9%)	20 (28.2%)	1 (4.0%)	0.01
IVF pregnancy	13 (13.5%)	7 (9.9%)	6 (24.0%)	0.09
MRI examination (weeks)	30.4 ± 3.4	30.2 ± 3.6	30.8 ± 2.8	0.42
Less than 26	10 (10.4%)	9 (12.7%)	1 (4.0%)	0.45
27–29	26 (27.1%)	20 (28.2%)	6 (24.0%)	0.80
30–32	29 (30.2%)	19 (26.8%)	10 (40.0%)	0.31
33 or more	31 (32.3%)	23 (32.3%)	8 (32.0%)	1.0
Location of placenta				
Anterior	25 (26.0%)	24 (33.8%)	1 (4.0%)	0.0030
Central	30 (31.3%)	25 (35.2%)	5 (20.0%)	0.21
Posterior	41 (42.7%)	22 (31.0%)	19 (76.0%)	0.00013
GA at delivery(weeks)	35.5 ± 2.57 (19–39)	35.4 ± 0.5 (19–39)	35.9 ± 1.78 (29–38)	0.39
Less than 27	1 (1.0%)	1 (1.4%)	0 (0%)	1
27–30	1 (1.0%)	1 (1.4%)	0 (0%)	1
31–35	42 (43.8%)	32 (45.1%)	10 (40.0%)	0.82
36 or more	52 (54.2%)	37 (52.1%)	15 (60.0%)	0.64
Operative procedure				
Cesarean hysterectomy	34 (35.4%)	30 (45.1%)	4 (16.0%)	0.027
CD	62 (64.5%)	41 (54.9%)	21 (84.0%)	
Histopathological diagnosis	34			
Placenta accreta	12 (12.5%)	9 (12.6%)	3 (12.0%)	1.0
Placenta increta	12 (12.5%)	11 (15.5%)	1 (4.0%)	0.175
Placenta percreta	7 (7.3%)	7 (9.9%)	0	0.184
Non-invasive placenta	2 (2.1%)	2 (2.8%)	0	1.0
Not performed	1 (1%)	1 (1.4%)	0	1.0

Abbreviations: CD, cesarean delivery; GA, gestational age; IVF, in vitro fertilization; MRI, magnetic resonance imaging. ^a^ Calculated as weight in kilograms divided by the square of height in meters, ^b^ A case of abortion was excluded.

**Table 2 jcm-08-02141-t002:** Accuracy of cervical canal angle for the diagnosis of posterior uterine wall adhesion.

Definition of Abnormal Cervical Angle		Sensitivity (%)	Specificity (%)	Positive Predictive Value	Negative Predictive Value
When a cervical canal angle of ≤10° is an abnormal sign	All patients (*n* = 96)	81.0	89.3	0.68	0.94
	MRI at a gestational age of ≤32 weeks (*n* = 65)	92.0	86.7	0.76	0.96
When a cervical canal angle of ≤0° is an abnormal sign	All patients (*n* = 96)	57.0	97.0	0.86	0.89
	MRI at a gestational age of ≤32 weeks (*n* = 65)	67.0	100	1.00	0.91

Upper panel lists the sensitivity, specificity, PPV, and NPV, when a cervical canal angle of ≤10° is designated as an abnormal sign. Lower panel lists the sensitivity, specificity, PPV, and NPV, when a cervical canal angle of ≤0° is designated as an abnormal sign. Abbreviations: PPV, positive predictive value; NPV, negative predictive value.

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
