# Peer review of "Horizontal Cervix as a Novel Sign for Predicting Adhesions on the Posterior Extrauterine Wall in Cases of Placenta Previa"

_jcm, 2019, doi:10.3390/jcm8122141_

Round 1
Reviewer 1 Report
This study was about predicting adhesions on the posterior extrauterine wall in placenta previa cases with the direction of the cervix on the MRI images, and very interesting work. This reviewer agrees with the author’s suggestion that these results might be useful in clinical application. However, some issue seems to be considered as the followings;
These data lacked longitudinal analysis according to gestational age and analyzed each case at various gestational age. So, the changes in the cervical canal angle during pregnancy according to gestational age could be predictive inappropriate with these data. Is there any effect on the cervical canal angle with birth weight, or twin pregnancy? Isn’t there any bias about uterine pathologies such as myoma, or adenomyosis? Is there any data related to the severity of extrauterine adhesion? If any, it could be useful information.
Author Response
Response to Reviewer 1 Comments
Reviewer 1
This study was about predicting adhesions on the posterior extrauterine wall in placenta previa cases with the direction of the cervix on the MRI images, and very interesting work. This reviewer agrees with the author’s suggestion that these results might be useful in clinical application. However, some issue seems to be considered as the followings;
Point 1
These data lacked longitudinal analysis according to gestational age and analyzed each case at various gestational age. So, the changes in the cervical canal angle during pregnancy according to gestational age could be predictive inappropriate with these data.
Thank you for your comments. I completely agree with your opinion. Due to the high cost of MRI, several cervical canal angles at different time periods cannot be determined. As you pointed out, our data may be biased; thus, this information has been added to the limitation in line 295. To improve the limitation, the cervical canal angle can be measured using a transvaginal ultrasonography at various gestational ages to investigate changes in the cervical canal angle during pregnancy in a future study. We have added these comments in line 311.
Point 2
Is there any effect on the cervical canal angle with birth weight, or twin pregnancy? Isn’t there any bias about uterine pathologies such as myoma, or adenomyosis? Is there any data related to the severity of extrauterine adhesion? If any, it could be useful information.
As you indicated, we should investigate the association among the cervical canal angle, birth weight, twin pregnancy, myoma, and adenomyosis. Unfortunately, we did not have enough data to investigate them; thus, this information has also been added in lines 283 and 300.

Reviewer 2 Report
This is a well written original paper and extensive effort being put in.
However, it’s hard for me to find out that how the information will help clinically to the clinicians or to do things differently after knowing this information by MRI.
This topic is good for research point of view but day to day routine clinical benefit is hard to be justified. I don’t think that large I international community will adopt it as an important information giving sign while managing cases of adherent placentas.
Author Response
Response to Reviewer 2 Comments
Reviewer 2
Point 1
This is a well written original paper and extensive effort being put in.
However, it’s hard for me to find out that how the information will help clinically to the clinicians or to do things differently after knowing this information by MRI.
We appreciate your insightful comments. Indeed, we could not show the surgical strategy when the positive sign of horizontal cervix was found. Therefore, whether identifying the horizontal cervix improves the surgical outcome should also be determined. We have added this information in line 307.
This topic is good for research point of view but day to day routine clinical benefit is hard to be justified. I don’t think that large I international community will adopt it as an important information giving sign while managing cases of adherent placentas.
We definitely agree with your opinion. The value of our finding may vary by county because the cost of MRI varied between $300 and $3000. In Japan, MRI costs about $300; thus, we think our findings are useful. However, if the cost is $3000, our findings may not be very useful. To improve the limitations of our study, further studies investigating the cervical canal through a transvaginal ultrasonography should be conducted. We have added the comments in line 311.

Reviewer 3 Report
Dear colleagues,
To even improve the quality of the paper Box&Whisker Plots should be created and compared to the normal group adjusted for gestational age.

Author Response
Response to Reviewer 3 Comments
Reviewer 3
Dear colleagues,
Point 1
To even improve the quality of the paper Box&Whisker Plots should be created and compared to the normal group adjusted for gestational age.
I recreated box and whisker plots for your presentations. Please cerate delta-values to compare the angles for the whole population adjusting for gestational age using box/whisker plots.
As per your suggestion, we tried to make Box & Whisker Plots to compare placenta previa with and without placenta accreta spectrum (PAS) in each gestational week (≥26 weeks, 27–30 weeks, 31–33 weeks, and ≤34 weeks). In this classification, only few cases were found to have placenta previa without PAS in the ≥26-week group and placenta previa with PAS in the ≤34-week group. Therefore, the cervical canal angle was compared between placenta previa with and without PAS in groups 2 and 3 (27–33 weeks). Unfortunately, MRI was performed in significantly higher gestational weeks in the non-PAS group than in the PAS group. To adjust the gestational age at MRI examination, we have revised the classification into the following four groups according to gestational age at the time of MRI examination: group 1 (≤26 weeks), group 2 (between 27 and 29 weeks), group 3 (30–32 weeks), and group 4 (≥33 weeks). Table 1 has also been revised based on this information. In this new classification, the cervical canal angle was compared between the PAS and non-PAS groups in groups 2 and 3. No significant change of gestational weeks at MRI examination and cervical canal angle was observed between the PAS and non-PAS groups. Based on these results, we considered that t the presence of PAS was not associated with the cervical canal angle. We have added the comments in line 163.
Point 2
In the paper the group 1 (<=26 wks) is described as >= 26wks. Please correct that.
This has been revised accordingly in line 71.
